# Prevalence and sociodemographic correlates of anogenital Human Papillomavirus (HPV) carriage in a cross-sectional, multi-ethnic, community-based Asian male population

**Su Pei Khoo[1], Mohd Khairul Anwar Shafii[1], Nirmala Bhoo-Pathy[1], Siew Hwei Yap[1], Shridevi Subramaniam[2], Nazrila Hairizan Nasir[3], Zhang Lin[4], Jerome Belinson[5], Pik Pin Goh[2], Xinfeng Qu[5], Patti Gravitt[6], Yin Ling Woo**[1] *

**1** Faculty of Medicine, University of Malaya, Kuala Lumpur, Malaysia, **2** National Clinical Research Centre, Ministry of Health, Putrajaya, Malaysia, **3** Division of Family Health Development, Ministry of Health, Putrajaya, Malaysia, **4** Beijing Genome Institute, Shen Zhen, China, **5** Preventive Oncology International Inc and the Cleveland Clinic, Cleveland, Ohio, United States of America, **6** Department of Epidemiology and Public Health, University of Maryland School of Medicine, Baltimore, Maryland, United States of America

* ylwoo@ummc.edu.my

## Abstract

### Background

Addressing the burden of HPV-associated diseases among men is increasingly becoming a public health issue. The main objective of this study was to determine HPV prevalence among a healthy community-based Malaysian men.

### Method

This was a cross-sectional study that recruited 503 healthy males from 3 community-based clinics in Selangor, Malaysia. Genital and anal samples were collected from each participant for 14 high risk and 2 low risk HPV DNA detection and genotyping. All participants responded to a set of detailed sociodemographic and sexual behaviour questionnaire.

### Results

The median age at enrolment was 40 years old (IQR: 31–50). The anogenital HPV6/11 prevalence was 3.2% whereas high risk HPV prevalence was 27.1%. The genital HPV prevalence for HPV6/11 was 2.9% while high risk HPV was 18.8%. HPV6/11 prevalence in the anal canal was 1.6% and high risk HPV was 12.7%. HPV 18 was the most prevalent genotype detected in the anogenital area. There was a significant independent association between genital and anal HPV infections.

### Conclusion

Anogenital HPV infection is common among Malaysian men. These findings emphasize the ubiquity of HPV infection and thus the value of population-wide access to HPV prevention.

**Data Availability Statement:** All relevant data are within the manuscript and its Supporting information files.

**Funding:** This study is supported by Merck Sharp & Dohme funding to YLW (https://www.msdmalaysia.com/home/). The funders had no role in study design, data collection and analysis, decision to publish, or preparation of the manuscript.

**Competing interests:** YLW has received travel grants and honoraria for speaking and participating at meetings by Merck Sharp & Dohme, Copan, Cepheid and Roche. YLW has also received investigatorinitiated study grants from Roche, Merck and Cepheid. This does not alter our adherence to PLOS ONE policies on sharing data and materials.

## Introduction

Human papillomavirus (HPV) infection is the most frequent sexually transmitted infection among men and women. HPV is classified into the high risk (HR) and low risk (LR) genotypes. The LR types including HPV 6 and 11 are the main cause of genital warts whereas HPV 16 and 18 are the more carcinogenic genotypes. HPV is also linked to approximately 50% of penile cancer, 88% of anal cancer and up to 56% of oropharyngeal cancer cases [1, 2]. While these cancers are most likely preventable with HPV vaccines, most of the HPV vaccination programmes are aimed at primary cervical cancer prevention [3]. The awareness and uptake HPV vaccines are low among men [4, 5]. Additionally, there is no approved test and recommendation for routine HPV screening among men [6]. The rate of cervical cancer has declined over years but the incidence rate of male HPV associated cancers have shown a significant increasing trend over the past decades [7–9]. In Malaysia, school-based national HPV vaccination program targeting 13-year-old girls was introduced in 2010. The uptake has been more than 80% since its introduction [10]. The high uptake of vaccination among Malaysian females might be beneficial to men as a result of herd immunity. To implement a strategic HPV-disease prevention program, it is therefore important to determine the baseline HPV prevalence among healthy Malaysian men.

## Materials and methods

### Study participants

This was a cross-sectional study that had recruited 503 healthy males via convenience sampling from September 2014 to February 2016. These participants were recruited from 3 community-based clinics in Selangor, Malaysia and most of them attended the clinics as a companion to their family members that required medical attention or routine check-ups. Participants aged 18 to 60 years that agreed to self/doctor's anogenital sampling were recruited for the study. Informed consent form was obtained from all participants. The exclusion criteria were acute illness or never having been sexually active. This study was approved by the Medical Research Ethics Committee (NMRR-13-444-14609), Ministry of Health Malaysia and the University of Malaya Medical Ethics Committee (MREC989.32).

### Anogenital sample and data collection

Anogenital samples were obtained using a brush (randomly selecting from either cytobrush from Hybribio Limited, Hong Kong, 'Just For Me' cervical sampler with courtesy of Preventive Oncology International, Hong Kong or Qiagen DNA Pap Cervical Sampler). Genital samples were collected by swabbing the shaft skin of penis for at least ten times. For anal sample collection, brush was inserted into the anus and turned for at least ten times. Self-sampling or doctor's sampling was performed according to participant's preference using the same sample collection method. All the brushes were then soaked in a tube containing 2ml of ThinPrep PreservCyt Solution (Hologic, USA) for preservation of cells during transportation. All participants responded to a set of questions on detailed sociodemographic and sexual behaviour data. The interview was carried out by a trained study staff member in private.

### HPV DNA detection and genotyping

All collected samples were kept at room temperature before extraction. Cells were pelleted before DNA extraction using DNeasy Blood & Tissue Kit (Qiagen, USA) as per manufacturer instructions within 2 weeks of sample collection. The final volume of extracted sample was 100ul. DNA samples were stored at -20 ℃ before transporting to Beijing, China for HPV

genotyping. HPV genotyping was performed using the BGISEQ-100 (Beijing Genome Institute (BGI)-assembled Ion Proton Sequencer from Life Technologies, South San Francisco, California, USA) as previously described [11]. Amplifications of 14 high risk (HR) HPV genotypes (16, 18, 31, 33, 35, 39, 45, 51, 52, 56, 58, 59, 66, and 68), 2 low risk (LR) HPV (6 and 11) and human β-globin (HBB) gene were done by a multiplexed polymerase chain reaction (PCR). The amplicons were then undergone magnetic bead purification, end repair reaction and adapter ligation. The HPV genotypes were detected in one sequencing ion chip. The sequences were mapped with the sample index and adapter sequences of the several adapted-indexed libraries. The reads per sample were then re-grouped and the detected HPV sequence were matched with a BGI-curated standard references HPV database from the National Center for Biotechnology Information (NCBI) by using the HPV typing software (China Food and Drug Administration (CFDA) registration number: 022702129, registered under Beijing Genome Institute). The sample was classified as HPV-positive if a particular HPV type reads and HBB were over a cut off value on a one-time testing basis. Samples with insufficient HBB DNA were considered invalid regardless of HPV reads.

## Statistical analysis

Demographic and sexual behaviour information such as age, age at sex debut, and number of lifetime sexual partner was recorded as continuous data. Marital status, ethnicity, highest attained education level, household income, circumcision status, smoking and alcohol drinking history were also recorded. Anogenital HPV positive infection was defined by detection of any HPV DNA genotypes at least one of the sites. HPV genotypes targeted by bivalent vaccine was defined as HPV16/18, those targeted by quadrivalent vaccine were HPV16/18/6/11 whereas nonavalent vaccine targeted genotypes were HPV16/18/6/11/31/33/45/52/58. The prevalence estimates were described using percentage. Association between HPV infection and variables were evaluated using Pearson Chi Square Test or Fisher Exact Test when 20% or more of the cell count were less than 5. Variable with p-value less than 0.2 in the univariable analysis and variables that were deemed important from the literatures were included in the multivariable analysis to determine the independent risk factor for HPV infection. A p-value of <0.05 was considered as statistically significant. All analyses were performed using IBM SPSS Statistics Version 20 and Microsoft Excel 2016.

## Results

### Study participants

A total of 503 healthy male subjects were enrolled in this study, 29 of them were excluded due to incomplete data/unable to provide samples (n = 474 available for analysis). The median age at enrolment was 40 years old (IQR: 31–50). The main ethnic groups were Malay (n = 248/474, 52.3%), followed by Chinese (n = 111/474, 23.4%) and Indian (n = 99, 20.9%). Most participants were heterosexual (n = 421/474, 88.8%), married (n = 329/474, 69.4%) while 72.4% (343/474) were smokers and 62.9% (297/472) reported to have been circumcised. The final analytic set with valid HPV results was not statistically different from the total enrolled population in this study.

### Sample analysis

A total of 474 anogenital samples were collected from the study population. However, 17.9% (85/474) of genital samples and 22.2% (105/474) of anal samples could not be tested due to insufficient amount of DNA, resulting in 389 genital samples and 369 anal samples with valid

HPV results for analysis. A total of 314 subjects provided both genital and anal samples with valid HPV results for analysis. There was a significant difference in term of sample sufficiency between self-collected samples and clinician-collected samples (p = 0.039 for genital samples and p<0.001 for anal samples, data shown in S1 Table). A significantly higher failure rate was detected among clinician collected samples (21.0% and 31.2%) compared to the self-collected samples (13.6% and 9.6%). Anal sample failure was also found to be highest among participants who used 'Just For Me' cervical sampler (31.3%, p<0.001) compared to Hybriobio cytobrush (23.9%) and Qiagen DNA Pap Cervical Sampler (5.2%), data shown in S2 Table.

## HPV prevalence and genotypes distribution

The overall anogenital HPV prevalence was 29.6% (93/314), with the highest HPV prevalence (32.1%) in the youngest age group (18–24) (Table 1). The anogenital HPV prevalence was also significantly higher among participants who had sex with men (47.8%, p = 0.04) (Table 2). The detected high risk (HR) HPV prevalence was 27.1% (85/314) whereas the prevalence of HPV6/11 was 3.2% (9/314) (Table 3). The prevalence of at least 1 HPV type at any anatomic site preventable by quadrivalent and nonavalent vaccines was 17.5% (55/314) and 22.0% (69/314), respectively. The genital HPV prevalence for the 16 types tested was 21.1% (82/389) whereas the HPV prevalence in the anal canal was 13.6% (50/369). Out of the 14 HR HPV genotypes being tested, HPV 18 was the most commonly detected type at both genital (6.2%, 24/389) and anal sites (6.0%, 22/369). This was followed by HPV 51 (3.1%, 12/389), HPV 52 (3.1%, 12/389) and HPV 16 (2.8%, 11/389) at the genital sites. For the anal canal, HPV 66 (2.7%, 10/369) was the second most common detected HPV genotype followed by HPV 45 (1.9%, 7/369) and HPV 51 (1.9%, 7/369). 19 out of 93 (20.4%) HPV positive individuals had concurrent anal and genital HPV infection (Fig 1).

## Correlates of prevalent HPV infection

The associations of HPV infection and sociobehavioural characteristics are shown in Tables 1 and 2 whereas the independent correlations analysed using Poisson Regression are shown in Table 4. Anogenital HPV infection does not vary across age groups (p = 0.952) and different races (p = 0.820) (Table 1). Smoking (p = 0.006) and alcohol drinker (p = 0.087) showed significant associations to anogenital HPV infection in the univariable analysis but was not found be independent risk factor to HPV infections. Age of sexual debut and history of having sex with men (MSM) correlated significantly with HPV infection in the univariable analysis (p = 0.044 and p = 0.040, respectively) but was not found to be independently associated with HPV infection in the multivariable analysis. However, there is a significant independent association between genital and anal HPV infection (p = <0.001).

## Discussion

To date, this is the first community-based HPV prevalence study among healthy men in Malaysia. The overall anogenital HPV prevalence for 14 high-risk (HR) genotypes and 2 low-risk (LR) wart-associated genotypes among healthy heterosexual male population in Malaysia was 29.6% with the highest prevalence demonstrated in the youngest age group (32.1%). Genital HPV prevalence was 21.1% and the HPV prevalence at the anal canal reported to be 13.6%. HPV18 was the most prevalent HPV genotype in both genital site and anal canal.

In a meta-analysis that included 40 publications on HPV detection in men, the prevalence of genital HPV infections was reported to be variable from 1.3% to 72.9% [12]. HR-HPV prevalence (18.8%) was comparable to male HR-HPV prevalence in studies from the United States, Denmark, and China (25.1–31.8%) [13–15]. The highest genital HPV prevalence was found to

**Table 1. HPV prevalence based on anatomic sites and demographics characteristics.**

| Variables | Anogenital Area | | | Genital Site | | | Anal Canal | | |
|---|---|---|---|---|---|---|---|---|---|
| | N (%) | HPV Prevalence (%) | p-value | N (%) | HPV Prevalence (%) | p-value | N (%) | HPV Prevalence (%) | p-value |
| **Overall** | 314 (100) | 29.6 | - | 389 (100) | 21.1 | - | 369 (100) | 13.6 | - |
| **Age (years)** | | | | | | | | | |
| 18–24 | 28 (8.9) | 32.1 | 0.952* | 36 (9.3) | 25.0 | 0.911* | 29 (7.9) | 17.2 | 0.289* |
| 25–30 | 43 (13.7) | 30.2 | | 55 (14.1) | 21.8 | | 56 (15.2) | 8.9 | |
| 31–40 | 90 (28.7) | 26.7 | | 111 (28.5) | 20.7 | | 108 (29.3) | 9.3 | |
| 41–50 | 78 (24.8) | 32.1 | | 93 (23.9) | 22.6 | | 92 (24.9) | 16.3 | |
| 51–60 | 75 (23.9) | 29.3 | | 94 (24.2) | 18.1 | | 84 (22.8) | 17.9 | |
| **Ethnicity** | | | | | | | | | |
| Malay | 170 (54.1) | 29.4 | 0.820* | 199 (51.2) | 21.1 | 0.905* | 204 (55.3) | 12.7 | 0.923* |
| Chinese | 78 (24.8) | 32.1 | | 91 (23.4) | 23.1 | | 90 (24.4) | 15.6 | |
| Indian | 55 (17.5) | 29.1 | | 86 (22.1) | 19.8 | | 62 (16.8) | 12.9 | |
| Others | 11 (3.5) | 18.2 | | 13 (3.3) | 15.4 | | 13 (3.5) | 15.4 | |
| **Education** | | | | | | | | | |
| None | 8 (2.5) | 12.5 | 0.678† | 10 (2.6) | 10.0 | 0.684† | 9 (2.4) | 0 | 0.223† |
| Primary | 44 (14.0) | 27.3 | | 62 (15.9) | 17.7 | | 49 (13.3) | 10.2 | |
| Secondary | 198 (63.1) | 32.3 | | 236 (60.7) | 23.7 | | 222 (60.2) | 16.2 | |
| Tertiary | 53 (16.9) | 26.4 | | 62 (15.9) | 17.7 | | 70 (19.0) | 12.9 | |
| Postgraduates | 11 (3.5) | 18.2 | | 19 (4.9) | 15.8 | | 19 (5.1) | 0 | |
| **Income (RM)** | | | | | | | | | |
| <1000 | 63 (20.1) | 34.9 | 0.452† | 76 (19.5) | 22.4 | 0.535* | 68 (18.4) | 16.2 | 0.833† |
| 1001–2000 | 109 (34.7) | 27.5 | | 137 (35.2) | 20.4 | | 124 (33.6) | 12.1 | |
| 2001–5000 | 116 (36.9) | 29.3 | | 143 (36.8) | 23.1 | | 138 (37.4) | 13.8 | |
| 50001–10000 | 20 (6.4) | 35.0 | | 24 (6.2) | 16.7 | | 31 (8.4) | 16.1 | |
| >10000 | 6 (1.9) | 0 | | 9 (2.3) | 0 | | 7 (1.9) | 0 | |
| Missing | - | - | | - | - | | 1 (0.3) | 0 | |
| **Smoking Status** | | | | | | | | | |
| No | 76 (24.2) | 42.1 | 0.006* | 99 (25.4) | 28.3 | 0.042* | 99 (26.8) | 16.2 | 0.375* |
| Yes | 238 (75.8) | 25.6 | | 290 (74.6) | 18.6 | | 270 (73.2) | 12.6 | |
| **Alcohol Status** | | | | | | | | | |
| No | 211 (67.2) | 26.5 | 0.087* | 253 (65.0) | 17.8 | 0.030* | 251 (68.0) | 13.9 | 0.747* |
| Yes | 103 (32.8) | 35.9 | | 136 (35.0) | 27.2 | | 118 (32.0) | 12.7 | |

*P-value generated using Pearson Chi Square test.

†P-value generated using Fisher exact test.

Missing data were not included in the analysis.

HPV: Human Papillomavirus

be in the youngest age group, similar to the female cervical HPV prevalence reported in the same population earlier [16]. However, compared to women, the male age-specific genital HPV prevalence curve was relatively flat. This results followed the global pattern which suggested similar HPV prevalence among the male population regardless of age [17, 18]. The most common genital HR-HPV genotypes (HPV 18, 51, 52, 16 and 45) reported in our study were similar to many other populations in the world [12, 14, 15, 18, 19]. This is important as these are the genotypes contributed to not only penile cancer but also cervical cancers [20–24]. Additionally, the common genotypes found in this study is similar to the one found among the women in the same setting [16].

**Table 2. Sexual characteristics and correlates of HPV infections.**

| Variables | Anogenital Area | | | Genital Site | | | Anal Canal | | |
|---|---|---|---|---|---|---|---|---|---|
| | N (%) | HPV Prevalence (%) | p-value | N (%) | HPV Prevalence (%) | p-value | N (%) | HPV Prevalence (%) | p-value |
| **Overall** | 314 (100) | 29.6 | - | 389 (100) | 21.1 | - | 369 (100) | 13.6 | - |
| **Circumcision Status** | | | | | | | | | |
| No | 116 (36.9) | 29.3 | 0.927* | 153 (39.3) | 20.3 | 0.649† | 125 (33.9) | 12.0 | 0.524* |
| Yes | 198 (63.1) | 29.8 | | 233 (59.9) | 21.3 | | 243 (65.8) | 14.4 | |
| **Don't know** | 0 | 0 | | 3 (0.8) | 33.3 | | 0 | 0 | |
| **Missing** | - | - | | - | - | | 1 (0.3) | 0 | |
| **Age of Sexual Debut (years)** | | | | | | | | | |
| <18 | 63 (20.1) | 33.3 | 0.044* | 76 (19.5) | 25.0 | 0.027* | 70 (19.0) | 10.0 | 0.119* |
| 18–25 | 161 (51.3) | 33.5 | | 205 (52.7) | 23.9 | | 187 (50.7) | 17.1 | |
| 26–30 | 52 (16.5) | 13.5 | | 60 (15.4) | 6.7 | | 65 (17.6) | 6.2 | |
| >30 | 35 (11.1) | 31.4 | | 44 (11.3) | 22.7 | | 43 (11.7) | 14.0 | |
| **Missing** | 3 (1.0) | 0 | | 4 (1.0) | 0 | | 4 (1.1) | 25.0 | |
| **Lifetime Female Sexual Partner** | | | | | | | | | |
| 1 | 95 (30.3) | 22.1 | 0.205* | 118 (30.3) | 12.7 | 0.012* | 122 (33.1) | 12.3 | 0.916* |
| 2 | 29 (9.2) | 24.1 | | 34 (8.7) | 11.8 | | 32 (8.7) | 15.6 | |
| 3+ | 154 (49.0) | 33.8 | | 189 (48.6) | 27.0 | | 172 (46.6) | 14.5 | |
| **Don't know/refused to answer** | 26 (8.3) | 34.6 | | 17 (4.4) | 23.5 | | 33 (8.9) | 12.1 | |
| **Missing** | 10 (3.2) | 40.0 | | 31 (8.0) | 25.8 | | 10 (2.7) | 10.0 | |
| **Current Female Sexual Partner** | | | | | | | | | |
| No | 72 (22.9) | 36.1 | 0.212* | 91 (23.4) | 26.4 | 0.351† | 82 (22.2) | 15.9 | 0.774* |
| Yes | 217 (69.1) | 26.7 | | 266 (68.4) | 19.2 | | 257 (69.6) | 12.8 | |
| **Refused to answer** | 15 (4.8) | 40.0 | | 1 (0.3) | 0 | | 20 (5.4) | 15.0 | |
| **Missing** | 10 (3.2) | 30.0 | | 31 (8.0) | 22.6 | | 10 (2.7) | 10.0 | |
| **New Female Partner in the Last 12months** | | | | | | | | | |
| No | 257 (81.8) | 28.4 | 0.668* | 317 (81.5) | 19.6 | 0.127† | 302 (81.8) | 13.6 | 0.942† |
| Yes | 24 (7.6) | 33.3 | | 32 (8.2) | 34.4 | | 26 (7.0) | 15.4 | |
| **Refused to answer** | 16 (5.1) | 37.5 | | 1 (0.3) | 0 | | 22 (6.0) | 13.6 | |
| **Missing** | 17 (5.4) | 35.3 | | 39 (10.0) | 23.1 | | 19 (5.1) | 10.5 | |
| **History of Receiving Oral Sex from Female Partner** | | | | | | | | | |
| No | 100 (31.8) | 25.0 | 0.520* | 129 (33.2) | 17.1 | 0.279† | 120 (32.5) | 16.7 | 0.489* |
| Yes | 174 (55.4) | 30.5 | | 213 (54.8) | 22.5 | | 197 (53.4) | 12.2 | |
| **Can't remember/refused to answer** | 20 (6.4) | 35.0 | | 6 (1.5) | 33.3 | | 27 (7.3) | 11.1 | |
| **Missing** | 20 (6.4) | 40.0 | | 41 (10.5) | 24.4 | | 25 (6.8) | 12.0 | |
| **History of Anal Sex with Female Partner** | | | | | | | | | |
| No | 266 (84.7) | 28.6 | 0.707* | 329 (84.6) | 20.7 | 0.739† | 314 (85.1) | 13.7 | 1.000* |
| Yes | 23 (7.3) | 34.8 | | 28 (7.2) | 21.4 | | 24 (6.5) | 12.5 | |
| **Can't remember/refused to answer** | 17 (5.4) | 35.3 | | 3 (0.8) | 33.3 | | 23 (6.2) | 13.0 | |
| **Missing** | 8 (2.5) | 37.5 | | 29 (7.5) | 24.1 | | 8 (2.2) | 12.5 | |

*(Continued)*

**Table 2.** (Continued)

| Variables | Anogenital Area | | | Genital Site | | | Anal Canal | | |
|---|---|---|---|---|---|---|---|---|---|
| | N (%) | HPV Prevalence (%) | p-value | N (%) | HPV Prevalence (%) | p-value | N (%) | HPV Prevalence (%) | p-value |
| **History of Having Sex with Men (MSM)** | | | | | | | | | |
| No | 276 (87.9) | 27.5 | 0.040* | 348 (89.5) | 19.3 | 0.011† | 324 (87.8) | 13.3 | 0.825† |
| Yes | 23 (7.3) | 47.8 | | 25 (6.4) | 44.0 | | 25 (6.8) | 16.0 | |
| Refused to answer | 0 | 0 | | 1 (0.3) | 0 | | 20 (5.4) | 15.0 | |
| Missing | 15 (4.8) | 40.0 | | 15 (3.9) | 26.7 | | - | - | |
| **Current Genital HPV Infections** | | | | | | | | | |
| No | - | | | - | | | 244 (66.1) | 9.4 | <0.001* |
| Yes | - | | | - | | | 70 (19.0) | 27.1 | |
| N/A | - | | | - | | | 55 (14.9) | 14.5 | |
| **Current Anal HPV Infections** | | | | | | | | | |
| No | - | | | 272 (69.9) | 18.8 | <0.001* | - | | |
| Yes | - | | | 42 (10.8) | 45.2 | | - | | |
| N/A | - | | | 75 (19.3) | 16.0 | | - | | |

*P-value generated using Pearson Chi Square test.

†P-value generated using Fisher exact test.

Missing data were not included in the analysis.

HPV: Human Papillomavirus

N/A: Not available

To date, there is limited data on the prevalence of anal HPV infections among healthy, heterosexual male populations. The HIM study that had included more than 1300 men from the United States, Brazil and Mexico had reported a comparable anal HPV prevalence (12.0%) in the heterosexual male population [25]. Another study done in China reported a lower anal HPV prevalence of 7.8% [26] compared to our study. Both studies reported a lower HR-HPV prevalence in the anal canal (7.0% and 6.5%, respectively) compared to our study (12.7%). This is most likely due to the smaller sample size of our study (n = 369) compared to these studies that had included a bigger study population (n = 1000–2000). However, we noted that the percentage reported in our study might be under or overreporting as samples failure was high (22.2%). More than half of the anal swab were collected via clinician sampling (58.2%) and clinician sampling showed a significant higher failure rate compared to self-sampling (p<0.001). Many studies had reported HPV 16, 6 and 18 as the most prevalent genotype found in anal HPV positive samples, similar to the findings of our study [27–30].

There is a significant independent correlation between genital HPV infection and anal HPV infection (p<0.001) in this study. 20.4% (19/93) of the HPV positive individuals had concurrent infection at genital area and anal canal and 63.2% (12/19) of them had the same HPV genotypes in both anal and genital sites. This finding is similar to a study by Pamnani et al (HIM Study) who reported a higher risk of acquiring anal HPV infection following genital HPV infection in a heterosexual male population, indicating a possibility of self-inoculation or other non-penetrative mechanism as the route of HPV transmission [31]. Another possible explanation to this finding is that men who have sex with men (MSM) were underreporting in this study. There is evidence from the HITCH study who found that majority of the HPV

**Table 3. Distribution of HPV genotypes in different anatomic sites.**

| HPV Genotype | HPV Prevalence (%, 95% Confidence Interval) | | |
|---|---|---|---|
| | Anogenital Area (n = 314) | Genital Site (n = 389) | Anal Canal (n = 369) |
| Any | 29.6 (25.5–33.7) | 21.1 (17.2–25.0) | 13.6 (10.1–17.1) |
| HR-HPV | 27.1 (23.1–31.1) | 18.8 (15.0–22.6) | 12.7 (9.3–16.1) |
| HPV 18 | 9.9 (7.2–12.6) | 6.2 (3.9–8.5) | 6.0 (3.5–8.3) |
| HPV 6 | 6.7 (4.4–9.0) | 3.9 (2.0–5.8) | 3.3 (1.5–5.1) |
| HPV 66 | 5.1 (3.1–7.1) | 1.8 (0.5–3.1) | 2.7 (1.0–4.4) |
| HPV 16 | 4.8 (2.9–6.7) | 2.8 (1.2–4.4) | 1.6 (0.3–2.9) |
| HPV 45 | 4.1 (2.3–5.9) | 2.3 (0.8–3.8) | 1.9 (0.5–3.3) |
| HPV 51 | 4.1 (2.3–5.9) | 3.1 (1.4–4.8) | 1.9 (0.5–3.3) |
| HPV 52 | 3.5 (1.8–5.2) | 3.1 (1.4–4.8) | 0 |
| HPV 11 | 1.0 (0.1–1.9) | 0.5 (-0.2–1.2) | 0.5 (-0.2–1.2) |
| HPV 31 | 1.0 (0.1–1.9) | 1.0 (0–2.0) | 0.3 (-0.3–0.9) |
| HPV 39 | 1.0 (0.1–1.9) | 1.0 (0–2.0) | 0 |
| HPV 58 | 1.0 (0.1–1.9) | 0.8 (-0.1–1.7) | 0.3 (-0.3–0.9) |
| HPV 59 | 1.0 (0.1–1.9) | 0.3 (-0.2–0.8) | 0.5 (-0.2–1.2) |
| HPV 68 | 1.0 (0.1–1.9) | 0.5 (-0.2–1.2) | 0.3 (-0.3–0.9) |
| HPV 56 | 0.6 (-0.1–1.3) | 0.5 (-0.2–1.2) | 0 |
| HPV 33 | 0.3 (-0.2–0.8) | 0.3 (-0.2–0.8) | 0 |
| HPV 6/11 | 3.2 (1.6–4.8) | 2.9 (1.3–4.5) | 1.6 (0.3–2.9) |
| HPV 16/18 | 14.3 (11.1–17.5) | 8.7 (6.0–11.4) | 7.6 (4.9–10.3) |
| HPV 6/11/ 16/18 | 17.5 (14.1–20.9) | 11.6 (8.5–14.7) | 9.2 (6.3–12.1) |
| HPV 6/11/16/18/31/ 33/45/ 52/58 | 22.0 (18.3–25.7) | 15.9 (12.4–19.4) | 10.0 (6.9–13.1) |

Data are presented using percentages with 95% confidence interval.

HPV: Human Papillomavirus

HR-HPV: High risk HPV types

transmission only occurs through sexual transmission [32]. In a recent statistics reported by the Ministry of Health Malaysia, there is a rise of sexually transmitted infections among men who have sex with men (MSM) over the decades in our population but most of the subjects had denied history of having sex with men (87.9%) in our study [33]. This is most likely due to the criminalisation of the homosexual behaviours in our setting.

Smoking and alcohol intake were reported to be the risk factors for HPV infection among men in the HIM study [34, 35]. The proposed biological pathway for this is the detrimental

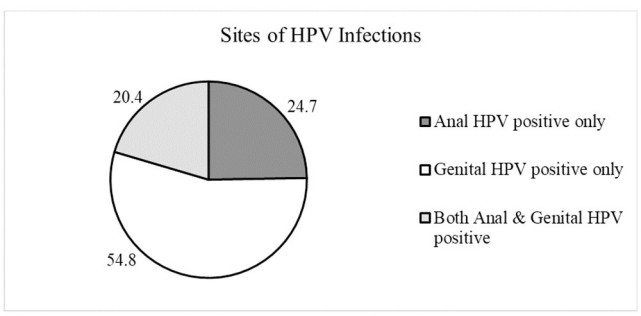

**Fig 1. Sites of HPV infection distribution among HPV-positive subjects (n = 93).**

**Table 4. Factors independently associated with any HPV genotype infection in various anatomic sites.**

| Variable | Anogenital Area (n = 284) | | | Genital Site (n = 278) | | | Anal Canal (n = 311) | | |
|---|---|---|---|---|---|---|---|---|---|
| | n | Adjusted prevalence ratio* (95% CI) | p-value | n | Adjusted prevalence ratio* (95% CI) | p-value | n | Adjusted prevalence ratio* (95% CI) | p-value |
| **Smoking Status** | | | | | | | | | |
| Yes | 219 | 1 | | 213 | 1 | | | - | |
| No | 65 | 1.35 (0.91 to 2.02) | 0.137 | 65 | 1.31 (0.81 to 2.13) | 0.270 | | - | |
| **Alcohol Status** | | | | | | | | | |
| No | 186 | 1 | | 185 | 1 | | | - | |
| Yes | 98 | 1.20 (0.82 to 1.76) | 0.340 | 93 | 1.37 (0.87 to 2.14) | 0.172 | | - | |
| **Age of Sexual Debut (years)** | | | | | | | | | |
| <18 | 63 | 1 | | 59 | 1 | | 63 | 1 | |
| 18–25 | 159 | 1.05 (0.69 to 1.60) | 0.825 | 156 | 0.88 (0.54 to 1.42) | 0.589 | 161 | 1.85 (0.81 to 4.23) | 0.142 |
| 26–30 | 51 | 0.46 (0.21 to 1.03) | 0.060 | 52 | 0.34 (0.12 to 1.01) | 0.051 | 52 | 1.01 (0.32 to 3.61) | 0.914 |
| >30 | 11 | 0.60 (0.15 to 2.45) | 0.476 | 11 | 0.77 (0.19 to 3.02) | 0.702 | 35 | 1.56 (0.53 to 4.57) | 0.425 |
| **Current Female Sexual Partner** | | | | | | | | | |
| No | 70 | 1 | | | - | | | - | |
| Yes | 214 | 0.84 (0.56 to 1.24) | 0.373 | | - | | | - | |
| **Lifetime Female Sexual Partner** | | | | | | | | | |
| 1 | 92 | 1 | | 94 | 1 | | | - | |
| 2 | 28 | 0.81 (0.39 to 1.71) | 0.581 | 26 | 0.49 (0.16 to 1.53) | 0.220 | | - | |
| 3+ | 154 | 0.97 (0.59 to 1.57) | 0.890 | 148 | 1.02 (0.55 to 1.90) | 0.959 | | - | |
| Don't know/ refused to answer | 10 | 0.86 (0.31 to 2.38) | 0.773 | 10 | 1.17 (0.44 to 3.14) | 0.756 | | - | |
| **New Female Partner in the Last 12months** | | | | | | | | | |
| No | | - | | 254 | 1 | | | - | |
| Yes | | - | | 24 | 0.95 (0.49 to 1.84) | 0.870 | | - | |
| **History of Having Sex with Men (MSM)** | | | | | | | | | |
| No | 264 | 1 | | 258 | 1 | | | - | |
| Yes | 20 | 1.41 (0.79 to 2.50) | 0.249 | 20 | 1.75 (0.96 to 3.17) | 0.066 | | - | |
| **Current Anal HPV Infections** | | | | | | | | | |
| No | | - | | 241 | 1 | | | - | |
| Yes | | - | | 37 | 2.45 (1.59 to 3.79) | **<0.001†** | | - | |
| **Current Genital HPV Infections** | | | | | | | | | |
| No | | - | | | - | | 241 | 1 | |
| Yes | | - | | | - | | 70 | 2.76 (1.59 to 4.81) | **<0.001¥** |

*P-value <0.05 is considered statistically significant and is marked in bold font.

†Derived using a multivariable Poisson regression model including smoking status, alcohol status, age at sexual debut, lifetime female sexual partner, new female partner in the last 12 months, history of having sex with men and current anal HPV infections as covariates.

¥Derived using a multivariable Poisson regression model including age at sexual debut and current genital HPV infections as covariates.

HPV, human papillomavirus.

effect of tobacco and alcohol on immunity which may lead to inefficient elimination of virus but more epidemiologic data are needed to confirm this hypothesis [34, 36, 37]. In study, we did not find an association between smoking and alcohol consumption to HPV acquisition. This finding was in consistent with other published evidence [38]. A plausible explanation is that increased risk of HPV infections among smokers and drinkers is often confounded by promiscuous sexual behaviours [38, 39]. Numerous studies reported early age of sexual debut and increased number of lifetime partner as significant risk factor for HPV infection among

me [25, 38, 40]. In our study, none of these sexual behavioural factors were found to be independently associated with HPV infection. This is most likely due to the small sample size of our study cohort.

There are some limitations in our study. By recruiting from only 3 clinics in Selangor and Kuala Lumpur, our sample may not adequately capture the variability of male HPV prevalence in Malaysia. Additionally, we also note the difference in term of racial composition in this study compared to the Malaysian population. As this is the first Malaysian study investigating HPV prevalence among healthy men, future studies with larger sample size and wider coverage should be conducted to verify the finding. We note that our data were comparable to international studies using population-based sampling. We do note a substantial rate of sample failure from both genital and anal swabs which might be resulting from inefficient sampling method as the sampling device was designed specifically for self-collected cervico-vaginal screening (not for genital/anal samples), and if this failure was not random according to HPV infection status may have led to biased prevalence estimates. As with all similar studies, measures of sociodemographic and sexual behaviours were self-reported and could be subjected to response desirability and recall bias.

In conclusion, the combined HR HPV and wart associated HPV6/11 prevalence among men in Malaysia was 29.6% (HR HPV prevalence alone 27.1%). These findings are comparable to other countries. The most prevalent HPV genotypes (18, 16, 45, 51 and 52) found in the male population reflect the genotype distribution in Malaysian women. While continuous high vaccine coverage in Malaysian women is likely to confer indirect positive impact on prevention of HPV infection and associated cancers in men through herd immunity, the data from this study will be important to support policymakers in Malaysia in optimization of their HPV elimination plans which will require balancing the cost-benefit of multiple HPV control strategies, such as gender-neutral vaccination and improved screening for the many cohorts of women infected prior to vaccine introduction.

## Supporting information

**S1 Table. Comparing sample sufficiency between clinician-collected samples and participant self-collected samples.**
(DOCX)

**S2 Table. Comparing sample sufficiency across different types of sampling device.**
(DOCX)

**S1 Data. Dataset for the study.**
(XLSX)

## Acknowledgments

The team is thankful to all individuals who participated in this study. The authors thank both Selangor State Health Department and Federal Territory of Kuala Lumpur Health Department for their cooperation throughout the recruitment process. We are also grateful to have assistance from the primary care clinics in recruiting subjects.

## Author Contributions

**Conceptualization:** Mohd Khairul Anwar Shafii, Nirmala Bhoo-Pathy, Nazrila Hairizan Nasir, Jerome Belinson, Pik Pin Goh, Patti Gravitt, Yin Ling Woo.

**Data curation:** Mohd Khairul Anwar Shafii, Nirmala Bhoo-Pathy, Patti Gravitt, Yin Ling Woo.

**Formal analysis:** Su Pei Khoo, Mohd Khairul Anwar Shafii, Patti Gravitt, Yin Ling Woo.

**Funding acquisition:** Nazrila Hairizan Nasir, Pik Pin Goh, Yin Ling Woo.

**Investigation:** Mohd Khairul Anwar Shafii, Siew Hwei Yap, Zhang Lin, Jerome Belinson, Xinfeng Qu, Yin Ling Woo.

**Methodology:** Mohd Khairul Anwar Shafii, Yin Ling Woo.

**Project administration:** Su Pei Khoo, Mohd Khairul Anwar Shafii, Siew Hwei Yap, Shridevi Subramaniam, Yin Ling Woo.

**Resources:** Nazrila Hairizan Nasir, Jerome Belinson, Pik Pin Goh, Xinfeng Qu, Yin Ling Woo.

**Supervision:** Nirmala Bhoo-Pathy, Yin Ling Woo.

**Validation:** Patti Gravitt, Yin Ling Woo.

**Visualization:** Yin Ling Woo.

**Writing – original draft:** Su Pei Khoo, Mohd Khairul Anwar Shafii, Yin Ling Woo.

**Writing – review & editing:** Mohd Khairul Anwar Shafii, Patti Gravitt, Yin Ling Woo.

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
