## [Decision Letter · Decision Letter 0]

3 Nov 2020

PONE-D-20-07025

Prevalence and Sociodemographic Correlates of Anogenital Human Papillomavirus (HPV) Carriage in A Cross-Sectional, Multi-Ethnic, Community-Based Asian Male Population

PLOS ONE

Dear Dr. Woo,

Thank you for submitting your manuscript to PLOS ONE. After careful consideration, we feel that it has merit but does not fully meet PLOS ONE’s publication criteria as it currently stands. Therefore, we invite you to submit a revised version of the manuscript that addresses the points raised during the review process.

We look forward to receiving your revised manuscript.

Kind regards,

Giovanni Delli Carpini

Academic Editor

PLOS ONE

Additional Editor Comments:

Even if the reported statistics is correct, the tables should be improved for better clarity; consider additional assistance by a medical statistician.

Journal Requirements:

"This study is funded by Merck Sharp & Dohme to YLW. The funders had no role in study design, data collection and analysis, decision to publish, or preparation of the manuscript.

URL of funder: " ext-link-type="uri" xlink:type="simple">https://www.msd-malaysia.com/home/"

We note that you received funding from a commercial source: Merck Sharp Dohme.

Reviewers' comments:

Reviewer's Responses to Questions

**Comments to the Author**

1. Is the manuscript technically sound, and do the data support the conclusions?

Reviewer #1: Partly

Reviewer #2: Yes

2. Has the statistical analysis been performed appropriately and rigorously? 

Reviewer #1: No

Reviewer #2: Yes

3. Have the authors made all data underlying the findings in their manuscript fully available?

Reviewer #1: Yes

Reviewer #2: Yes

4. Is the manuscript presented in an intelligible fashion and written in standard English?

Reviewer #1: Yes

Reviewer #2: Yes

5. Review Comments to the Author

Reviewer #1: This is an interesting and important study, yet I do have following concerns.

1. The representativeness of the sample: the authors recruited 503 males based on convenience sampling. Besides the sample size being small, the authors failed to discuss the representativeness of the sample; for example, Chinese ethnicity comprise about 24% of the sample: what’s the racial composition of the Malaysian population? Also, they were sampled from 3 hospitals; how representative of those 3 hospitals?

2. I have some issue with Table 4. Statistical significance was not achieved in most of the categories, which might be related with their small sample size (thus insufficient power)., Also, some categories have number of patients around 10 or 20, raising concerns regarding the reliability of the results.

Reviewer #2: The manuscript entitled: “Prevalence and sociodemographic correlates of anogenital human papillomavirus (HPV) carriage in a cross-sectional, multi-ethnic, community-based Asian male population”, analyze HPV prevalence among healthy Malaysian men and the association with sociodemographic features.

The work is interesting since a large cohort was analyzed. Healthy men were recruited from three clinics in Malaysia and 474 were available for the study. Samples were taken (self or doctor’s sampling) from the anogenital region, anal canal, and genitals; and a questionnaire with sociodemographic and sexual behavior data was applied. Samples were processed for DNA extraction and HPV genotyping with a kit that detects 14 high-risk and 2 low-risk HPV types. Finally, 389 genital samples and 369 anal samples had an HPV status result.

Within the relevant results is that:

-Clinician collected samples were less effective for DNA sufficiency than self-collected, showing a significant difference.

-The overall anogenital HPV prevalence was 29.6%, being 27% for high-risk types and 3% for low-risk, with the highest prevalence in the younger men's group. HPV18 was the most commonly detected type.

- From the HPV positive men 24.7 % had anal positivity only; 54.8 % had genital positivity only and 20.4% were positive for both genital and anal regions, having the 63% of them the same viral type. Therefore, a significant independent association between genital and anal HPV infection was found.

- HPV prevalence was higher among men having sex with men, although in the multivariable analysis an independent association was not found.

- Significant associations with HPV positivity were found for smoking or drinking, but were not found to be independent risk factors for HPV infection.

- No significant associations was found between HPV prevalence and the other

sociodemographic characteristics or sexual behavior analyzed, although it is observed that there is a lot of missing data, which can give an uncertain result.

The statistics presented in tables is well performed.

The discussion is appropriate where the results obtained in the literature concerning other populations are adequately compared. Limitations of the study are also stated.

Comments:

A figure legend is missing in Figure 1.

6. PLOS authors have the option to publish the peer review history of their article (what does this mean?). If published, this will include your full peer review and any attached files.

Reviewer #1: No

Reviewer #2: No

---

## [Author Response · Author response to Decision Letter 0]

27 Nov 2020

Additional Editor Comments:

Funding and Competing interest statements:

I confirm and approve the following:

Funding:

This study is supported by Merck Sharp Dohme funding to YLW (https://www.msd-malaysia.com/home/). The funders had no role in study design, data collection and analysis, decision to publish, or preparation of the manuscript.

Competing interests:

YLW has received travel grants and honoraria for speaking and participating at meetings by Merck Sharp Dohme, Copan, Cepheid and Roche. YLW has also received investigator-initiated study grants from Roche, Merck and Cepheid. This does not alter our adherence to PLOS ONE policies on sharing data and materials.

Even if the reported statistics is correct, the tables should be improved for better clarity; consider additional assistance by a medical statistician.

Thank you for your comment. All statistical analysis in the manuscript were reviewed and edited for a better presentation by a medical statistician.

Reviewer #1: 

This is an interesting and important study, yet I do have following concerns.

1. The representativeness of the sample: the authors recruited 503 males based on convenience sampling. Besides the sample size being small, the authors failed to discuss the representativeness of the sample; for example, Chinese ethnicity comprise about 24% of the sample: what’s the racial composition of the Malaysian population? Also, they were sampled from 3 hospitals; how representative of those 3 hospitals?

Thank you for your comment.

The current racial distribution in Malaysia are 69.6% of Malays, 22.6% of Chinese and 6.8% of Indians. (Reference: Department of Statistics Malaysia, 2020). Our study cohort comprised of 52.3% of Malay, 23.4% of Chinese and 20.9% of Indian. We note the differences of racial distribution in our study cohort as compared to the population therefore have added this point at the study limitation paragraph by adding this sentence below at Page 19, Line 257-258.

“Additionally, we also note the differences in the racial composition of the study cohort compared to the Malaysian population.”

This cohort was part of another larger study aiming to determine the HPV prevalence among Malaysian therefore the study cohort were sampled from these clinics. These clinics are located at urban and suburban areas of Selangor and Kuala Lumpur, which are among the most densely populated states in Malaysia. However, we agreed that these clinics could not adequately represent the whole Malaysian population. We emphasized this study limitation by including the following at Page 19, Line 255-260 as shown below:

“By recruiting from only 3 clinics in Selangor and Kuala Lumpur, our sample does not represent the male HPV prevalence in Malaysia. Additionally, we also note the differences in the racial composition of the study cohort compared to the Malaysian population. As this is the first Malaysian study investigating HPV prevalence among healthy men, future studies with larger sample size and wider coverage should be conducted to verify our findings.”

2. I have some issue with Table 4. Statistical significance was not achieved in most of the categories, which might be related with their small sample size (thus insufficient power)., Also, some categories have number of patients around 10 or 20, raising concerns regarding the reliability of the results.

Thank you for your comment. 

For your information, the sample size for this study was calculated using a formula by Daniel WW 1999 with an expected prevalence of 65% based on the HIM study’s finding (Giuliano AR et al. 2008) and margin error of 5%. The required sample size was 350.

We do note that statistical significance was not achieved in most of the categories and agree that this might be due to the small number in some of the subcategories. This was explained as a study limitation at Page 19 Line 254-259 as shown below:

“By recruiting from only 3 clinics in Selangor and Kuala Lumpur, our sample does not represent the male HPV prevalence in Malaysia. Additionally, we also note the differences in the racial composition of the study cohort compared to the Malaysian population. As this is the first Malaysian study investigating HPV prevalence among healthy men, future studies with larger sample size and wider coverage should be conducted to verify our findings.”

In addition, we have added a paragraph to discuss further on the findings in Table 4 at Page 19 Line 242-252, please kindly see below:

Smoking and alcohol intake were reported to be the risk factors for HPV infection among men in the HIM study [34,35]. The proposed biological pathway for this is the detrimental effect of tobacco and alcohol on immunity which may lead to inefficient elimination of virus but more epidemiologic data are needed to confirm this hypothesis [34,36,37]. In our study, we did not find an association between smoking and alcohol consumption to HPV acquisition. This finding was in consistent with other published evidence [38]. A plausible explanation is that increased risk of HPV infections among smokers and drinkers is often confounded by promiscuous sexual behaviours [38,39]. Numerous studies reported early age of sexual debut and increased number of lifetime partner as significant risk factor for HPV infection among men [25,38,40]. In our study, none of these sexual behavioural factors were found to be independently associated with HPV infection. This is most likely due to the small sample size of our study cohort. 

Reviewer #2: 

1. The manuscript entitled: “Prevalence and sociodemographic correlates of anogenital human papillomavirus (HPV) carriage in a cross-sectional, multi-ethnic, community-based Asian male population”, analyze HPV prevalence among healthy Malaysian men and the association with sociodemographic features.

The work is interesting since a large cohort was analyzed. Healthy men were recruited from three clinics in Malaysia and 474 were available for the study. Samples were taken (self or doctor’s sampling) from the anogenital region, anal canal, and genitals; and a questionnaire with sociodemographic and sexual behavior data was applied. Samples were processed for DNA extraction and HPV genotyping with a kit that detects 14 high-risk and 2 low-risk HPV types. Finally, 389 genital samples and 369 anal samples had an HPV status result.

Within the relevant results is that:

-Clinician collected samples were less effective for DNA sufficiency than self-collected, showing a significant difference.

-The overall anogenital HPV prevalence was 29.6%, being 27% for high-risk types and 3% for low-risk, with the highest prevalence in the younger men's group. HPV18 was the most commonly detected type.

- From the HPV positive men 24.7 % had anal positivity only; 54.8 % had genital positivity only and 20.4% were positive for both genital and anal regions, having the 63% of them the same viral type. Therefore, a significant independent association between genital and anal HPV infection was found.

- HPV prevalence was higher among men having sex with men, although in the multivariable analysis an independent association was not found.

- Significant associations with HPV positivity were found for smoking or drinking, but were not found to be independent risk factors for HPV infection.

- No significant associations was found between HPV prevalence and the other

sociodemographic characteristics or sexual behavior analyzed, although it is observed that there is a lot of missing data, which can give an uncertain result.

The statistics presented in tables is well performed.

The discussion is appropriate where the results obtained in the literature concerning other populations are adequately compared. Limitations of the study are also stated.

Thank you for your comments.

We have added a paragraph in discussion to further discuss on the associations between sociodemographic and sexual behavioural factors (results in Table 4). Please refer to Page 19, Line 242-252 as shown below:

Smoking and alcohol intake were reported to be the risk factors for HPV infection among men in the HIM study [34,35]. The proposed biological pathway for this is the detrimental effect of tobacco and alcohol on immunity which may lead to inefficient elimination of virus but more epidemiologic data are needed to confirm this hypothesis [34,36,37]. In our study, we did not find an association between smoking and alcohol consumption to HPV acquisition. This finding was in consistent with other published evidence [38]. A plausible explanation is that increased risk of HPV infections among smokers and drinkers is often confounded by promiscuous sexual behaviours [38,39]. Numerous studies reported early age of sexual debut and increased number of lifetime partner as significant risk factor for HPV infection among men [25,38,40]. In our study, none of these sexual behavioural factors were found to be independently associated with HPV infection. This is most likely due to the small sample size of our study cohort. 

We understand the concern regarding the reliability of the finding due to some missing data and small number in some of the categories. Therefore, we have edited the study limitations to clarify at Page 19, Line 254-259 as shown below:

“By recruiting from only 3 clinics in Selangor and Kuala Lumpur, our sample does not represent the male HPV prevalence in Malaysia. Additionally, we also note the differences in the racial composition of the study cohort compared to the Malaysian population. As this is the first Malaysian study investigating HPV prevalence among healthy men, future studies with larger sample size and wider coverage should be conducted to verify our findings.”

2. Comments:

A figure legend is missing in Figure 1.

Thank you for pointing this out.

The figure legend is added in Figure 1. 

Fig 1. Sites of HPV infection distribution among HPV-positive subjects (n=93). 20.4% of the HPV-positive subjects had concurrent genital and anal HPV infections whereas more than half of them (54.8%) had only genital HPV infection.

---

## [Decision Letter · Decision Letter 1]

7 Jan 2021

Prevalence and Sociodemographic Correlates of Anogenital Human Papillomavirus (HPV) Carriage in A Cross-Sectional, Multi-Ethnic, Community-Based Asian Male Population

PONE-D-20-07025R1

Dear Dr. Woo,

We’re pleased to inform you that your manuscript has been judged scientifically suitable for publication and will be formally accepted for publication once it meets all outstanding technical requirements.

Kind regards,

Giovanni Delli Carpini

Academic Editor

PLOS ONE

Additional Editor Comments (optional):

Reviewers' comments:

Reviewer's Responses to Questions

**Comments to the Author**

1. If the authors have adequately addressed your comments raised in a previous round of review and you feel that this manuscript is now acceptable for publication, you may indicate that here to bypass the “Comments to the Author” section, enter your conflict of interest statement in the “Confidential to Editor” section, and submit your "Accept" recommendation.

Reviewer #2: All comments have been addressed

2. Is the manuscript technically sound, and do the data support the conclusions?

Reviewer #2: Yes

3. Has the statistical analysis been performed appropriately and rigorously? 

Reviewer #2: I Don't Know

4. Have the authors made all data underlying the findings in their manuscript fully available?

Reviewer #2: Yes

5. Is the manuscript presented in an intelligible fashion and written in standard English?

Reviewer #2: Yes

6. Review Comments to the Author

Reviewer #2: I consider that the authors responded adequately to the comments and the manuscript is now suitable for publication.

7. PLOS authors have the option to publish the peer review history of their article (what does this mean?). If published, this will include your full peer review and any attached files.

Reviewer #2: No

---

## [Editor Report · Acceptance letter]

11 Jan 2021

PONE-D-20-07025R1 

Prevalence and Sociodemographic Correlates of Anogenital Human Papillomavirus (HPV) Carriage in A Cross-Sectional, Multi-Ethnic, Community-Based Asian Male Population 

Dear Dr. Woo:

I'm pleased to inform you that your manuscript has been deemed suitable for publication in PLOS ONE. Congratulations! Your manuscript is now with our production department. 

Kind regards, 

on behalf of

Dr. Giovanni Delli Carpini 

Academic Editor

PLOS ONE